# Municipal Executive Recommendation by Citizens: Who Is Most Significant?

**Galvão Meirinhos** [1], **Maximino Bessa** [2], **Carmem Leal** [3], **Márcio Sol** [4], **Amélia Carvalho** [5] and **Rui Silva** [3,*]

1   Department of Literacy, Arts and Communication, University of Trás-os-Montes and Alto Douro—UTAD, 5000-801 Vila Real, Portugal; gsm@utad.pt
2   Department of Engineering, University of Trás-os-Montes and Alto Douro—UTAD, 5001-801 Vila Real, Portugal; maxbessa@utad.pt
3   CETRAD Research Center, University of Trás-os-Montes and Alto Douro—UTAD, 5001-801 Vila Real, Portugal; cleal@utad.pt
4   NECE-Research Center in Business Sciences, University of Beira Interior, 6201-001 Covilhã, Portugal; marcio.oliveira@ipleiria.pt
5   CIICESI, ESTG, Porto Polytechnic, 4200-465 Porto, Portugal; acarvalho@estg.ipp.pt
*   Correspondence: ruisilva@utad.pt

**Abstract:** This paper explores which variables are more significant in municipal executive recommendation by citizens. We estimated the influence of public dimensions, such as municipe loyalty, municipe satisfaction, and municipe perceived value in municipal executive recommendation by citizens. Then, we tried to understand if the citizen's opinions influenced the evaluation of the municipal executive recommendation. The parishes of the municipality of Valongo were selected and analyzed, namely the parishes of Alfena, Campo e Sobrado, Valongo, and Ermesinde, and a total of 998 questionnaires were collected. Data were collected in November 2020 in the different parishes under study. It was concluded that all studied dimensions were statistically significant in the final structural estimated model. The structural results point to municipe loyalty and municipe satisfaction dimensions having a direct, positive, and statistically significant influence on municipal executive recommendation. On the other side, the municipe perceived value dimension has a direct positive but not statistically significant influence on municipal executive recommendation. This study showed that a loyal and satisfied citizen recommends the continuity of the municipal executive in the city's political leadership in which he or she lives. Therefore, for the municipal executive administration, it is fundamental to know which dimensions the society considers most important in order to be able to remain in the management of the shared destinies of a city. In this sense, political decisions throughout the mandates can be directed, on the one hand, to the satisfaction and loyalty of the citizens and, on the other hand, to the balanced management of the destinies of this type of public entity.

**Keywords:** municipal executive recommendation; municipe loyalty; municipe satisfaction; municipe perceived value

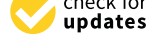



## 1. Introduction and Background

Public opinion has become a reality and a constructing variable of municipal recommendation. The sources of influence come in different forms, such as the citizen's perceived value, the citizen's satisfaction of the citizen, and the loyalty of the voter (Margolis and Mauser 1989). Therefore, understanding these sources is essential to adapt the agents' and institution's public action and policy. In this context, the quality of public services appears as a central element and a priority for municipal executives in order to meet public expectations. The satisfaction of citizens is closely linked to quality, a theoretical territory where conceptual discrepancies still exist (Martin et al. 2020; Charantimath 2011). Most experts define the concept of quality in general terms as conformity (Martin et al. 2020), suitability

(Charantimath 2011), and customer satisfaction (Sukaris and Yulia 2021; Alghamdi 2016). The evaluation of citizen satisfaction is an integrated and holistic exercise in which public institutions must listen to and involve citizens in their activities in order to better meet their personal needs and expectations. In this systematised and controlled interaction, public institutions improve their performance in the provision of public services. In this sense, municipal recommendation is directly related to satisfaction, loyalty, and favourable public opinion (Meirinhos et al. 2022).

With regard to citizen satisfaction, factors, such as the efficient provision of services and the provision of high-quality services at the lowest possible cost, are mentioned as two aspects in which citizens report having the greatest dissatisfaction (Kushner and Siegel 2005). Moreover, the relationship between internal management performance and citizen satisfaction in the public sector is seen as a predominant factor affecting satisfaction, with a positive correlation between management performance and citizen satisfaction, in addition to citizen satisfaction helping to trigger improvements in management practices in the managers of public organisations who wish to improve citizen satisfaction with the specific service offered by their organisation (Kim and Lee 2012). On the other hand, security issues are key to increasing citizen satisfaction, with police presence being a common significant predictor of citizen satisfaction (Bouranta et al. 2015). Moreover, with regard to the political ideology of citizens, there is a relationship between the variation of political choices on offer in a party system and citizen satisfaction, suggesting that when party choices in a political system are closer ideologically to the average position of voters in terms of left-right, overall citizen satisfaction increases (Ezrow and Xezonakis 2011). In relation to citizens' expectations, it is concluded that citizens' expectations, and especially disconfirmation of expectations—factors that have not previously been considered in empirical studies on the determinants of citizen satisfaction—play a key role in forming satisfaction judgements regarding the quality of urban services, suggesting that urban managers should seek to promote not only high-quality services, but also high expectations among citizens (Van Ryzin 2004).

Citizen satisfaction is a popular means of performance management, as citizens are customers who care about the quality of public goods and services and about equity and accessibility to public goods and services. In this sense, it is necessary that public managers develop and employ skills that recognise the complex consumerist and democratic attributes of citizens in a public economy (Collins et al. 2019). Still, with regard to aspects of environmental sustainability, it was found that air pollution, which is more than an environmental or health issue, has an influence on the likelihood of citizens being satisfied with environmental administration. In this sense, it has been found that air pollution erodes citizens' satisfaction and presents political costs promoted by citizens' dissatisfaction where this issue occurs (Chen et al. 2020). Moreover, the various ethnic groups report different levels of satisfaction regarding the communication process within the local public administration, with a significant difference between the perceptions and expectations of the citizens regarding the communication process within the municipal councils. The results of this study show that in the responsiveness dimension of the quality of communication, satisfaction is significantly affected by the respondents' belonging to an ethnic group. In this context, in public policy making, local public sector managers should recognize the importance of ethnic characteristics and give weight to the ethnic characteristics of the community members they serve (Bente 2014).

As citizen satisfaction is a central theme in evaluating the performance of public services, it should be analysed over time to implement an effective public action programme, having upstream the aggregates of the dimension of the citizen's perceived value and downstream the voter's loyalty. In the context of this work, and in conceptual terms, we define the following:

- Citizen's perceived value: The overall perception of value ascertained by the difference between the expected value in the form of benefits and the total cost for the citizen in the form of personal and/or financial sacrifices (Chang and Dibb 2012). The citizen's

perceived value is formed by the set of mental representations, judgements, and meanings stored around functional, symbolic, emotional, and social realities.

- Citizen satisfaction: The result of a comparison process between performances and a standard (Rico et al. 2022; Gendel-Guterman and Billig 2021). The interaction between the parties allows the citizen to judge the action of the public entity (in the form of performance) against the citizen's legitimate expectations. When expectations are equal to or even exceeded, this creates and sediments the satisfaction of the citizen, where the result repeated over time consolidates the perception of satisfaction.
- Voter loyalty: The predisposition guiding a certain attitude (Tommasetti et al. 2018) or voting Behaviour (Zins 2001). The hypothetical behaviour focuses on past experience, whereas the affective dimension in the form of attitudes is based on future actions. It should be kept in mind that, similar to commercial loyalty, political loyalty means the repetition not of the purchase but of the repeated vote for a party or public or political personality (Venturino and Seddone 2020). Other authors, such as Zeithaml et al. (1996), add as a demonstration of loyalty the indicator of recommendation to third parties.

Recommendation has become one of the main marketing tools. There is no better source for electoral promotion than the recommendation coming from a satisfied citizen, creating institutional value in several ways. The act of advising and referring has gained space in personal decisions where the influence is real (Ejimabo 2015) and very recently leveraged by social networks (Kim et al. 2020; Xevelonakis 2016). Recommendation has become a means of increasing the mental pregnancy of the value of products and services, to the detriment of the intrinsic qualities of the offer itself (Kumar et al. 2010). In this context, companies and organizations are increasingly interested in identifying the agents with strength and capacity of recommendation in order to manage and leverage the marketing, financial, political and social value. Thus, recommendations have become free and silent acts of promotion that increasingly assume a strategic value in the public image and reputation of companies, organisations, and public institutions. Much of this knowledge stems from works related to customer value, e.g., Customer Referral Value by V. Kumar, Andrew Petersen, and Robert Leone (Kumar et al. 2007, 2010) and regarding the present and future customer performance value, Customer Life Time Value by Rajkumar Venkatesan (Venkatesan and Kumar 2004). Based on this, we can by plasticity understand that the public recommendation of a citizen is directly dependent on the quality of the public service offer and the satisfaction arising from his or her relationship with the municipality. Another important work in this field is by Kumar et al. (2016), who present customer value not only in terms of financial and performance value, but also the effective influence on other customers or potential customers. This paper also coined the term "customer perceived value", which was formulated as the understanding between objective product attributes and customer perceived attributes.

The theoretical body of the relationship marketing area clearly indicates the need to define a relationship model capable of boosting the recommendation processes (Palmatier et al. 2006). In fact, as early as 1957, the works of R.C. Brooks confirmed a process of mutual influence between customers (Brooks 1957), whose interpersonal relationship influences customer choice (Menzel and Katz 1955; Dichter 1966). Nowadays, the processes of influence and interpersonal relationships are growing at an unprecedented rate, which allows us to foresee the increase in the importance of recommendation as a management area. Other authors suggest that the higher the level of satisfaction, the higher the probability of recommendation, while the opposite situation is also plausible (Neto et al. 2005). Therefore, and for similarity, we can state that the enchantment effect by the mayor or the municipal executive triggers a recommendation dynamic, and in the opposite case a regret effect in case the citizen has voted for the elected municipal team, hence the importance of developing a serious relationship model between the municipality and the community, where high levels of satisfaction boost this multiplier effect of positive references about people and the institution. In this way, and taking into account the historical references that deal with

the subject, our approach is multifaceted and is able to assess the sources and variables that influence and contribute to municipal recommendation. Our theoretical model is global, integrative, structural, longitudinal, probabilistic, and estimates the performance of the municipality based on the expectations and perceptions of the citizens, crossing this reality with the achievements of the municipal executive. This model thus makes it possible to observe the nature of the public recommendation according to the levels of the citizen's perceived value, the citizen's satisfaction and the voter's loyalty, in a quantifiable, comparable and relatable perspective.

The choice of these research variables (Appendix A) is because, in the literature, they are the most addressed and the most influential on citizen satisfaction with local and national public administration. As citizens currently demand and expect more and better services from the public sector, the response must be fast and efficient to speed up processes, facilitate access and provide high-quality standards. The choice of these variables (municipe loyalty, executive recommendation, municipe satisfaction and municipe perceived value) is the most used method in this type of quantitative research, of descriptive character, being the target of study the citizens who use the services provided by the councils. These variables show sufficient robustness to evaluate the overall satisfaction of the citizen and, as a consequence, the recommendation of the municipal executive.

These variables allow us to analyse a model based on the premise that excellent results in organisational performance, citizens/customers, people, and society are achieved through a driving strategy of leadership and planning, people, partnerships, resources and processes, looking at the organisation from different angles at the same time; a holistic approach to the analysis of the organisation's performance.

## 2. Method

### 2.1. Study Background

Our overall model consists of four dimensions. First, in the scope of municipe satisfaction, we have indicators such as satisfaction with the different intervention axes (formed by 13 variables); satisfaction with municipal services (formed by 20 variables); satisfaction with the municipal executive (formed by 4 variables); and overall satisfaction with the municipality (formed by 1 variable) in a total of 38 variables. Regarding organizational performance and perceived quality (27 variables), contestation and complaint of the municipal executive (formed by 4 variables) and notoriety, image, and reputation (20 variables). In the indicator notoriety, image, and reputation, a series of statements were presented in which the respondents positioned themselves, namely statements that are related to seriousness, credibility, accountability, sensitivity, transparency, innovation, creativity, trust, public visibility, media strength, public communication management, and politics. This aggregate allows us to ascertain the mental representation residing in the collective memory about a set of vectors related to the knowledge, representation, and social evaluation of elected officials' quality of political action.

### 2.2. Data Collection Questionnaire

The data collection questionnaire was based on the theoretical and validated models: SERVQUAL, the Common Measurement Tool (CMT), the Common Assessment Framework (CAF), Speyer's model, and the European Customer Satisfaction Index (ECSI). However, it should be noted that none of these models has been adjusted and validated for public administration or the evaluation of Portuguese public services. Therefore, our model and respective survey are a reference in understanding the influence of dimensions, such as municipe loyalty (ML), municipe satisfaction (MS), and municipe perceived value (MPV) dimensions, having a direct positive and statistically significant influence on municipal executive recommendation (MER).

In this sense, this paper aims to explore the relationships among variables and determine the dependence among dimensions, indicators, and variables present in the general model of municipal executive recommendation.

According to the literature review, it is possible to hypothesize that the ML, MS and MPV dimensions increase municipal executive recommendation (MER). In this sense, the following research hypotheses were defined in Figure 1:

**H1:** *Municipe loyalty (ML) has a positive effect on municipal executive recommendation (MER).*

**H2:** *Municipe satisfaction (MS) has a positive effect on municipal executive recommendation (MER).*

**H3:** *Municipe perceived value (MPV) has a positive effect on municipal executive recommendation (MER).*

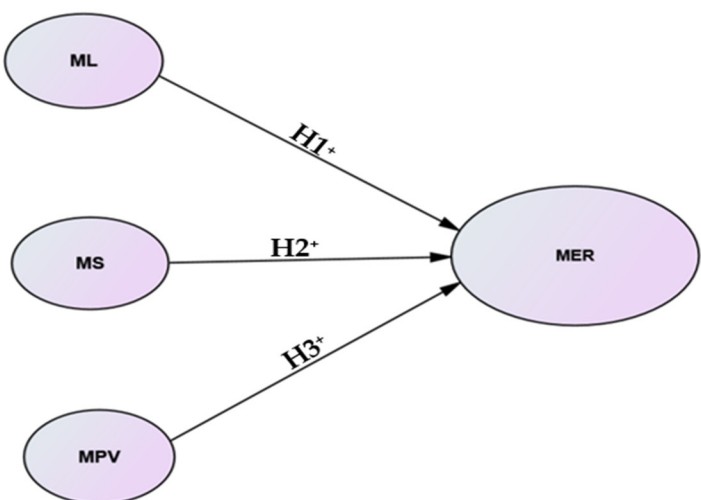

**Figure 1.** Research model.

*2.3. Data Collection Tools and Sample*

The data collection instrument used was scientifically tested and validated, due to the various levels of complexity and specialization in its construction. The construction work is based on a macro-structure, namely the information to be provided to respondents; the flow and sequence of questions; the appearance of the survey; the survey sections, and the use of filters and control questions. On the other hand, the micro-structure of the construction of the survey has to do with the response scales and the respective assertive statistical treatment. Another concern was related to the language used in the survey, given the weight of the population over 50 years of age in most Portuguese municipalities. In addition to the obstacles at the linguistic level, there are the inevitable differences in meanings for the same signifier, between two or more interlocutors, which can enhance error and variability in the interpretation of utterances. However, in addition to formal validation, there is also content validity that measures the degree of understanding of the questions by respondents. It is clear that the survey, like any other research instrument, has its advantages and disadvantages. In our opinion, the advantages far outweigh the disadvantages, because the survey can be used simultaneously with a large number of subjects spread over vast regions. In addition, people may feel more secure about the anonymity of responses and, as a result, express more freely the responses they consider more personal. In this way, a whole series of protocol and methodological concerns were taken into account to avoid biases or even the halo effect (Ghiglione et al. 2001), which is why certain more global questions are posed at the end of each section of the survey.

The quantitative scientific method presents itself as an instrument of knowledge acquisition provided by collecting, classifying, analysing, and interpreting data collected through the face-to-face questionnaire by approaching citizens in the different parishes of the municipality under study. In this way, a consistent and coherent survey was developed with the very particular reality of Portuguese municipalities, having as references the following theoretical models: SERVQUAL developed by (Parasuraman et al. 1993); Common

Measurement Tool (CMT) of the Canadian Management Center (Strickland, (Canada), and for Management Development. Learning Centre 1998); Common Assessment Framework (CAF) inspired by the Excellence Model of the European Foundation for Quality Management (European Foundation for Quality Management or EFQM) (Engel 2002) and the Speyer Model of the Deutschen Universität für Verwaltungswissen-schaften (Speyer 2012; and the European Customer Satisfaction Index (ECSI) (Ciavolino and Dahlgaard 2007), based on the American Consumer Satisfaction Index of the University of Michigan. The questionnaire is divided into three dimensions, with four indicators for each dimension, totalling 127 closed-ended questions associated with estimating the model's dimensions, indicators, and variables, which were tested through a pilot survey before the actual data collection. A 5-point Likert-type numerical scale was used in the survey, with the extreme points presenting the semantic description "Strongly disagree" and "Strongly agree", respectively. It is important to say that the scale is composed of a set of statements with a logical or empirical relationship and is a form of evaluation aimed at measuring a concept or a characteristic of the individual. At the end of the survey, and on an optional basis, the respondent is informed of the possibility of being part of a panel of citizens to be set up to obtain their responses electronically every six months. If the citizen accepts to participate in the panel, all his data are processed in accordance with the General Data Protection Regulation (RGPD). On the other hand, this nominative and electronic contact information is used exclusively to allow you to have private access to the questionnaire area, excluding any possibility of relating your personal data to the nature of the survey responses.

The participants were citizens between 18 and 82 years old, of which 48.3% were male and 51.7% were female. A total of 2260 questionnaires were collected (Table 1).

**Table 1.** Characterisation of the sample.

| Towns | Total Population | Sample | Male | Female |
|---|---|---|---|---|
| Alfena | 18,125 | 504 | 198 | 306 |
| UF Campo e Sobrado | 15,969 | 403 | 227 | 176 |
| Ermesindde | 38,798 | 711 | 322 | 389 |
| Valongo | 25,920 | 642 | 344 | 298 |
| | | **2260** | **1091** | **1169** |

The sample was collected randomly on the streets of each city. People were approached about their willingness to answer a questionnaire evaluating the work of the municipal executive. Five enumerators were assigned to each city, who, through paper questionnaires or the use of devices (IPAD/TABLET), allowed respondents to give their anonymous opinion on municipe loyalty (ML), municipe satisfaction (MS), and municipe perceived value (MPV) and municipal executive recommendation (MER). The sample represents about 3% of the total population since the total population of each municipality is 98,812 inhabitants. Although the robustness of the sample does not reach 5% of the population, the sample collected from men and women is balanced and allows relevant conclusions to be drawn.

### 2.4. Confirmatory Factorial Analysis (CFA)

To perform the AFC, a model was tested with a set of variables corresponding to the six constructs under analysis and another one with the removal of variables whose factorial loadings were lower than 0.5 (having chosen the one whose adjustment of variables revealed better statistical consistency (Brown 2006; Marôco 2010). We can verify the results of the final model estimation, which was improved by removing variables belonging to the constructs MS, OPPQ, and NIR. The most statistically significant model presented the following statistical evidence, ($\chi^2/df$ = 2.314, RMSEA = 0.086, SRMR = 0.0322, NFI = 0.896, GFI = 0.902, AGFI = 0.917 and CFI = 0.950), after removing some items that made it statistically more robust, having excluded all items whose factor loadings were less than 0.3 (Marôco 2010; Hair et al. 2010). Regarding the reliability of the items and factors, the sample

obtained, consisting of 998 Valongo municipe citizens, shows a good internal consistency ($\alpha = 0.961$).

## 3. Results

In Table 2, it is possible to see a summary of the hypotheses that were tested, using what was found to be the best research model, as well as the results that were obtained, which allow one to conclude that such dimensions account for the variation that occurs in MER as ML ($\beta = 0.900$, $p < 0.001$) and MS ($\beta = 0.080$, $p < 0.05$), and MPV ($\beta = 0.030$, $p > 0.05$). Only two of three dimensions were statistically significant in the final structural estimated model (Figure 2). The structural results point to ML and MS dimensions having a direct, positive, and statistically significant influence on MER, validating research hypotheses H1 and H2. Conversely, the MPV dimension has a direct positive but not statistically significant influence on NIR, not validating research Hypothesis 3.

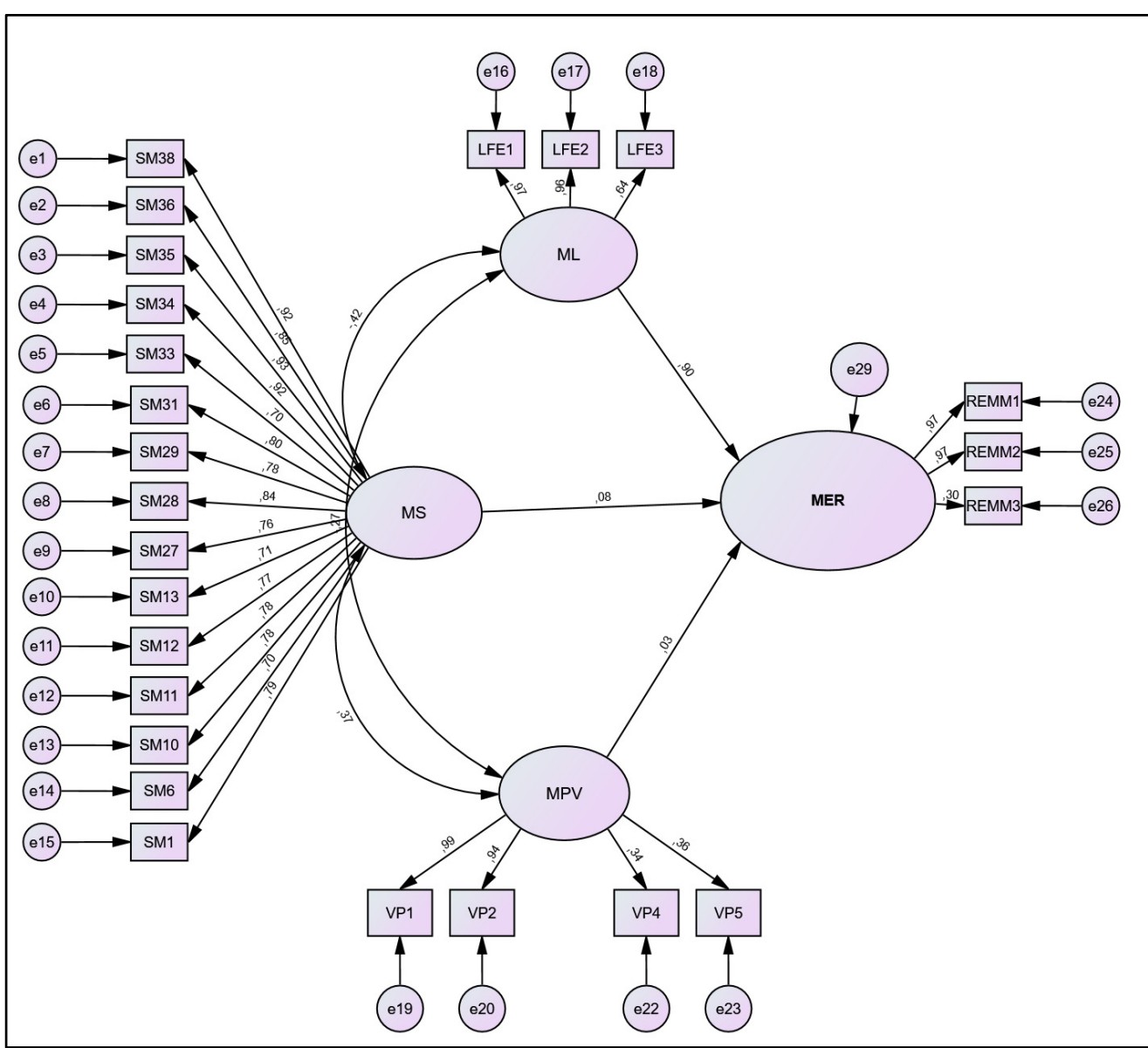

**Figure 2.** Final research model.

**Table 2.** Research hypotheses and statistical results.

| Hypotheses | Relation | Regression Coefficient | Standard Error | t | *p*-Value | Result |
|---|---|---|---|---|---|---|
| **H1** | ML → MER | 0.900 | 0.032 | 27.346 | <0.001 | Supported |
| **H2** | MS → MER | 0.080 | 0.033 | 2.722 | <0.05 | Supported |
| **H3** | MPV → MER | 0.030 | 0.023 | 1.163 | >0.05 | Not Supported |

After the validity and reliability of the final model (MER), some variables whose scores were less than 0.3 were removed, thus improving the model's internal consistency.

The results also conclude that the dimensions affecting municipal executive recommendation were MS and CCME. It should be noted that the OPPQ dimension had a positive impact on NIR, which did not affect it statistically significantly.

It has been noted that dimensions ML and MS proved to be the most pertinent dimensions concerning MER increase. ML was the strongest MER predictor, followed by MS, with a slight regression factor.

The tested hypothesis model accounted for most of the variance of dependent variables. In addition, most variables were highly correlated, strongly affecting NIR.

## 4. Discussion and Conclusions

This scientific article is the result of a major research project that aimed to gauge the opinions of municipal residents on several dimensions that affect their daily lives as members of constituency communities belonging to the cities of Alfena, Campo, Sobrado, Valongo, and Ermesinde belonging to the district of Porto. These peripheral towns are very important as they are densely populated. Moreover, the opinion of these citizens is crucial in the electoral decision of these cities, hence the importance of carrying out this study in the second largest district of Portugal.

The research model tested showed that two dimensions (municipal loyalty and municipal satisfaction) positively influence the municipal executive recommendation of the citizens of these cities.

The validation of Hypotheses 1 and 2 of this study allowed us to understand that municipe loyalty and municipe satisfaction. These results corroborate previous studies which stated that loyal citizens were more likely to re-elect the same municipal executive to represent them in public policies and decisions that affect their lives. In this sense, it can be seen that citizens currently demand and expect more and better services from the public sector, so the response must be fast and efficient in order to speed up processes, facilitate access and provide high quality standards. The satisfaction and loyalty of the citizen is intrinsically linked to the evolution of the very concepts of quality of public services, so that the perceived quality leads to an increase in interest and support for the work of the municipal executive. Therefore, the quality of these public services made available to the citizens will lead the population to support the re-election of the same municipal executive in a future electoral vote. In fact, the citizen, through the payment of his taxes, supports the functioning of the public administration, and as such, it is expected that when he has to resort to public services, he demands quality, assuming intrinsically that this can increase his satisfaction and loyalty.

In short, citizen satisfaction can be translated into loyalty, which represents the intention or predisposition to support the executive that publicly manages and administers the population's interests. This satisfaction and loyalty a later expressed in the electoral act, which is a pivotal occasion for demonstrating and experiencing democracy, where individuals and groups can express their satisfaction or dissatisfaction, their needs or aspirations, the desire for stability or even the will for change.

In turn, the dimension municipal perceived value had no impact on the municipal executive recommendation, thus not validating hypothesis 3. Even though the municipality's perceived value is a global perception of value ascertained by the difference between the expected value in the form of benefits and the total cost for the citizen in the form of

personal and/or financial sacrifices, we consider that it should have some impact on the Municipal Executive Recommendation. However, the results obtained are in line with what has been historically stated, confirming that only with high levels of satisfaction does a recommendation take place. Therefore, the dimension of the citizen's perceived value is related to the knowledge of valuing the attributes, which, as a stage prior to satisfaction, makes sense that it does not have enough strength to trigger the reality of the recommendation.

This scientific research results from a project implemented in Northern Portugal and analyses the main and most relevant results generated and obtained through the use of a computer platform called CIDIUS®, developed by the authors of this work, which aims to support the decision-making process of Portuguese mayors. This study showed that a loyal and satisfied citizen recommends the continuity of the municipal executive in the city's political leadership in which he or she lives. Therefore, for the municipal executive administration, it is fundamental to know which dimensions the society considers most important in order to be able to remain in the management of the shared destinies of a city. In this sense, political decisions throughout the mandates can be directed, on the one hand, to the satisfaction and loyalty of the citizens and, on the other hand, to the balanced management of the destinies of this type of public entity.

## 5. Limitations, Contributions and Implications of the Study

Data collection instruments can always be questioned because others have been previously validated in the literature, which allows for obtaining better and different results in other contexts and cultures. Perhaps one of the limitations of this study is the use of collection instruments adapted to the reality of municipalities, and this study was conducted in some cities of the Porto district. Moreover, the robustness of the sample, which does not reach 5%, is a limitation to be improved in future studies. However, these cities are densely populated and have the most important social and political impact on municipal executives. Therefore, the outstanding contribution of this study is the direct analysis of the local public reality, listening to those who are honestly concerned (citizens) in improving the quality of their living conditions.

This study showed that a loyal and satisfied citizen recommends the continuity of the municipal executive in the city's political leadership in which he or she lives. Therefore, for the municipal executive administration, it is fundamental to know which dimensions the society considers most important in order to be able to remain in the management of the shared destinies of a city. In this sense, political decisions throughout the mandates can be directed, on the one hand, to the satisfaction and loyalty of the citizens and, on the other hand, to the balanced management of the destinies of this type of public entity.

Studies such as these may, in the future, be replicated in other areas of the country and extrapolated internationally to other socio-cultural realities. Maybe, in the future, these kinds of studies can be performed more frequently for the municipal executive to have periodic information about their public and political job.

**Author Contributions:** Conceptualization, G.M., M.S., A.C. and R.S.; methodology, R.S.; software, R.S.; validation, G.M., M.B. and C.L.; formal analysis, R.S.; investigation, G.M.; resources, C.L. data curation, R.S.; writing—original draft preparation, G.M.; writing—review and editing, C.L.; visualization, M.B.; supervision, G.M., M.S., A.C. and R.S.; project administration, G.M., M.S., A.C. and R.S.; funding acquisition, G.M., M.B., C.L. and R.S. All authors have read and agreed to the published version of the manuscript.

**Funding:** The work of author Rui Silva is supported by national funds, through the FCT—Portuguese Foundation for Science and Technology under the project UIDB/04011/2022 and by NECE-UBI, Research Centre for Business Sciences, Research Centre under the project UIDB/04630/2022. The work of author Amélia Carvalho is supported by National funds, in CIICESI through the FCT—Portuguese Foundation for Science and Technology under the project UIDB/04728/2020.

**Institutional Review Board Statement:** The study was conducted in accordance with the Declaration of Helsinki and approved by the Institutional Review Board.

**Informed Consent Statement:** Informed consent was obtained from all subjects involved in the study.

**Data Availability Statement:** Not applicable.

**Acknowledgments:** The authors gratefully acknowledge the University of Trás-os-Montes and Alto Douro and CETRAD (Centre for Transdisciplinary Development Studies) and University of Beira Interior (NECE-UBI) and Porto Polytechnic-CIICESI-ESTG.

**Conflicts of Interest:** The authors declare no conflict of interest.

## Appendix A

| Survey Variables | Study Dimension |
|---|---|
| I would vote for the current Mayor.<br>I would vote for the members of the current Municipal Executive.<br>I would vote for the current Mayor. | **Municipe Loyalty** |
| Environment: revitalising public spaces, rationalising natural resources, preserving the environment, air quality.<br>Culture and leisure: congresses, museums, events, shows, festivals.<br>Sports: existence of sports and leisure equipment, promotion of (school) sports activity, physical activity as occupation of free time/school holidays.<br>Economy and entrepreneurship: development of economic activity in the municipality, business/industrial parks, business incubators, science and technology parks.<br>Education: school network (management of school groupings of the various cycles, school meals).<br>Urban hygiene: urban cleaning, cleaning of public buildings and equipment.<br>Internationalisation: international promotion of the municipality, sports and business activities.<br>Social intervention: childhood, youth, old age, employment, training, precariousness, economy, health.<br>Youth: management of youth centres, cultural promotion.<br>Mobility and transport: mobility/public transport network, other mobility (cycle paths), parking, heavy traffic induction roads, road safety.<br>Safety: road prevention and control, safety and monitoring of the elderly, policing (environment, commercial and public space control, surveillance and vigilance).<br>Tourism: information offices, tourist routes.<br>Urbanism: public space management, public heritage maintenance, licensing, infrastructures.<br>Is satisfied with the notoriety, image and reputation of the Municipality.<br>Is satisfied with the attention that the Municipality gives to the citizens.<br>Is he/she is satisfied with the functioning of the municipal services.<br>He/she is satisfied with the fees charged by the Municipality.<br>Overall, he/she is satisfied with the way the Municipality works. | **Municipe Satisfaction** |
| The fees charged are in line with the economic reality of the citizen.<br>The quality of the services justifies the fees charged by the Municipality.<br>The payment process is simple and fast.<br>The means of payment of the fees are adequate for the citizen.<br>Payment deadlines are reasonable and adjusted to the citizen. | **Municipe Perceived Value** |
| I would recommend the Mayor.<br>I would recommend the members of the current Municipal Executive.<br>I would advise this municipality to live and visit.<br>Would recommend this municipality for investment.<br>You intend to keep your place of residence in the municipality. | **Municipal Executive Recommendation** |

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
