# Peer review of "Municipal Executive Recommendation by Citizens: Who Is Most Significant?"

_admsci, doi:10.3390/admsci12030081_

Round 1
Reviewer 1 Report
Dear authors the topic is interesting, in addition I find the validation model used in this study very strong and clear.
But is necessary to improve some aspects as:
the introduction section with more previous relevant studies. Because one of the weakness point of this paper is the absence of a literature section. In order to fill this gap is necessary to improve the introduction.
To give a better quality of this paper is necessary to explain better the choice of the variables and the different hyphotesis connected with the research questions.
Materials and methods section: is necessary to add a table with the
explanation of the variables used in the analysis.
Discussion and Conclusion: One part of this section is in Spanish
Author Response
Dear authors the topic is interesting, in addition I find the validation model used in this study very strong and clear.
R:\ Dear Reviewer, thanks for your great job and enforces to give me ideas and good advice to improve the paper.Many thanks for all your job in this paper review.
But is necessary to improve some aspects as:
the introduction section with more previous relevant studies. Because one of the weakness point of this paper is the absence of a literature section. In order to fill this gap is necessary to improve the introduction.
R:\Dear reviewer, we have consulted several papers on Web of Science and improved our introduction on the topic. We have chosen articles related to citizen satisfaction in different aspects.
To give a better quality of this paper is necessary to explain better the choice of the variables and the different hyphotesis connected with the research questions.
R:\We have inserted an extra explanation at the end of the introduction, after the hypotheses/research model.
Materials and methods section: is necessary to add a table with the
explanation of the variables used in the analysis.
R:\Dear reviewer, such a proposal makes perfect sense. We have added this table in appendix.
Discussion and Conclusion: One part of this section is in Spanish
R:\Our sincere apologies. It was a lapse in translation. thank you
Thank you and Best Regards
Reviewer 2 Report
The manuscript is a good, interesting and mature piece of research. The methodology is well-designed and is consistent with the objectives of the study. The interpretation and discussion of results are clear, objective and consistent. The conclusions summarize well the results obtained and are consistent with the work presented.
Below, you can find my minor remarks for further improving the manuscript:
I suggest to shift lines 115-138 to Methodology section.
Lines 195-205: how the authors choose citizents taking part in the survey? Where they were found and what was a key to compose a responding group? How many people live in the towns where the research was done? Are the samples representative?
The part of the text in Discussion and Conclusions section is not in English.
Author Response
The manuscript is a good, interesting and mature piece of research. The methodology is well-designed and is consistent with the objectives of the study. The interpretation and discussion of results are clear, objective and consistent. The conclusions summarize well the results obtained and are consistent with the work presented.
R:\
Dear Reviewer, thanks for your great job and enforces to give me ideas and good advice to improve the paper. Many thanks for all your job in this paper review.
Below, you can find my minor remarks for further improving the manuscript:
I suggest to shift lines 115-138 to Methodology section.
R:\ Done. We agree, makes more sense.
Lines 195-205: how the authors choose citizents taking part in the survey? Where they were found and what was a key to compose a responding group? How many people live in the towns where the research was done? Are the samples representative?
R:\ Very important awareness. Done.
The part of the text in Discussion and Conclusions section is not in English.
R:\ Our sincere apologies. It was a lapse in translation. Thank you and sorry about that.
Round 2
Reviewer 1 Report
Well done!
Author Response
Dear reviewer many thanks for all your job in this paper revision.
Best Regards and thanks for all